# Gingival solitary chemosensory cells are immune sentinels for periodontitis

Xin Zheng[1,2,3], Marco Tizzano[2,3], Kevin Redding[2], Jinzhi He [1], Xian Peng[1], Peihua Jiang[2], Xin Xu[1]*, Xuedong Zhou[1]* & Robert F. Margolskee [2]*

Solitary chemosensory cells (SCCs) are epithelial sentinels that utilize bitter Tas2r receptors and coupled taste transduction elements to detect pathogenic bacterial metabolites, triggering host defenses to control the infection. Here we report that SCCs are present in mouse gingival junctional epithelium, where they express several Tas2rs and the taste signaling components α-gustducin (Gnat3), TrpM5, and Plcβ2. Gnat3$^{-/-}$ mice have altered commensal oral microbiota and accelerated naturally occurring alveolar bone loss. In ligature-induced periodontitis, knockout of taste signaling molecules or genetic absence of gingival SCCs (gSCCs) increases the bacterial load, reduces bacterial diversity, and renders the microbiota more pathogenic, leading to greater alveolar bone loss. Topical treatment with bitter denatonium to activate gSCCs upregulates the expression of antimicrobial peptides and ameliorates ligature-induced periodontitis in wild-type but not in Gnat3$^{-/-}$ mice. We conclude that gSCCs may provide a promising target for treating periodontitis by harnessing innate immunity to regulate the oral microbiome.

[1] State Key Laboratory of Oral Diseases & National Clinical Research Center for Oral Diseases, Department of Cariology and Endodontics, West China Hospital of Stomatology, Sichuan University, 610041 Chengdu, China. [2] Monell Chemical Senses Center, Philadelphia, PA 19104, USA. [3] These authors contributed equally: Xin Zheng, Marco Tizzano. *email: xin.xu@scu.edu.cn; zhouxd@scu.edu.cn; rmargolskee@monell.org

Periodontitis, a bacterially induced chronic inflammation of the tooth-supporting tissue (i.e., gingiva, periodontal ligament, and alveolar bone), is the sixth most prevalent infectious disease and the most common cause of tooth loss worldwide[1–3]. Recent studies have shown that periodontitis results from polymicrobial dysbiosis, which perturbs the ecologically balanced oral microbiota underlying normal periodontal homeostasis[4–8]. The host innate immune system is highly active in healthy periodontal tissue; however, an imbalance or disruption in innate immunity also contributes to the destruction of periodontal tissue[1,9–11]. The mechanisms underlying the complex host-microbiota interactions that determine periodontal homeostasis remain poorly defined, with very limited knowledge of the specific host receptors that detect pathogenic oral bacteria and/or their metabolites.

Pattern-recognition receptors, such as Toll-like receptors and Nod-like receptors, are reported to be involved in host–bacterial interactions in periodontal tissue[1,10,12–16]. Recent studies in other tissues identified taste-cell-like solitary chemosensory cells (SCCs) as another means by which bacteria can evoke the host's innate immune defenses[17–23]. In mouse airways, SCCs detect Gram-negative bacterial quorum-sensing acyl-homoserine lactone (AHL) molecules through a canonical taste transduction cascade involving bitter taste receptors (Tas2rs) and downstream taste signaling elements, including the G protein subunit α-gustducin (Gnat3), phospholipase C beta 2 (Plcβ2), and transient receptor potential cation channel melanostatin 5 (TrpM5)[17,19,23]. The activation of nasal SCCs triggers protective respiratory reflexes and inflammatory/immune responses to prevent damage to the epithelium and avoid the danger[22–24]. In addition, the activation of bitter TAS2R taste receptors expressed in human nasal SCCs stimulates mucosal secretion of antimicrobial peptides (AMPs) that repress the growth of respiratory pathogens[20]. In the gut, tuft or brush cells (a type of SCC)[25,26] detect and evoke innate immune responses against helminthic parasites[18]. SCCs serve as innate immune sentinels in a variety of anatomical locations, including but not limited to gastric mucosa and biliary tract[27], tracheal and laryngeal glandular ducts[28], and urinary tract[17].

Here we identify and characterize SCCs in mouse gingival junctional epithelium. Gingival SCCs (gSCCs) may respond to bacterial signals via their Tas2rs and downstream taste signaling components to trigger host innate immune responses to prevent overgrowth of oral bacteria. The gSCCs regulate the oral microbial composition and protect against periodontitis.

## Results

**Taste signaling elements are expressed in mouse gingiva.** To determine whether SCCs are present in gingiva, we examined the expression of mRNAs and proteins for taste signaling components. Reverse transcription polymerase chain reaction (RT-PCR) revealed expression in mouse periodontal tissue of 10 out of 35 mouse Tas2r bitter taste receptors (Tas2r105, Tas2r108, Tas2r118, Tas2r119, Tas2r126, Tas2r134, Tas2r135, Tas2r137, Tas2r138, and Tas2r143), along with taste signaling elements α-gustducin, TrpM5, and Plcβ2 (Fig. 1a; see also Supplementary Fig. 1A). In addition, G protein-coupled receptors Gpr41 and Gpr43 that detect short-chain fatty acids[25] and free fatty acid receptors Grp120 and CD36 were found in the gingival tissue (Supplementary Fig. 1B).

Immunohistochemistry of gingival tissue identified cells expressing α-gustducin in the sulcular and junctional epithelium but not in the marginal epithelium (Fig. 1b). Double labeling showed that these α-gustducin+ cells also expressed Plcβ2 (Fig. 1b), a defining feature of SCCs in other tissues. Using TrpM5-GFP transgenic mice, we found that α-gustducin-

immunoreactive cells also expressed TrpM5-driven green fluorescent protein (GFP; Supplementary Fig. 1C). Altogether, the gingival cells expressing α-gustducin, Plcβ2, and TrpM5 are likely to be SCCs. Immunohistochemistry of gingiva from Pou2f3−/− mice, which lack SCCs in all tissues examined[29], found no α-gustducin+/Plcβ2+ gSCCs (Fig. 1b). In contrast, immunohistochemistry of Gnat3−/− mice indicates that, while they lack this G protein subunit (Fig. 1b), they still retain SCCs (based on immunoreactivity to Plcβ2; Supplementary Fig. 1C). A hallmark of SCCs in many other tissues is the use of acetylcholine as a downstream effector[17,22,27,28]. ChAT-GFP mice, expressing GFP from the promoter of the acetylcholine-synthesizing enzyme choline acetyltransferase (ChAT), showed a population of GFP+ cells in the marginal epithelium that did not overlap with the α-gustducin+ populations in the junctional epithelium, indicating that gSCCs are not cholinergic (Supplementary Fig. 1C).

**gSCCs utilize Tas2rs to respond to bacterial molecules.** AHLs, Gram-negative bacterial quorum-sensing molecules, are detected by SCCs in mouse nasal epithelium[23] and by two types of human nasal chemosensory cells: ciliated cells via the human bitter taste receptor TAS2R38 and SCCs by bitter taste receptors other than TAS2R38[20,30,31]. To determine whether any of the 10 Tas2rs expressed in gSCCs could be activated by AHLs, we heterologously expressed the receptors in HEK293 cells along with the chimeric G protein Gα16gust44 to couple receptor activation to $Ca^{2+}$ mobilization. Of the 10 Tas2rs tested, only Tas2r105 elicited $Ca^{2+}$ responses to the bacterially produced 3-oxo-C12-homoserine lactone (HSL) (purified from *Escherichia coli* transfected with the LasI construct; see Supplementary Table 1) and to two synthetic HSLs: 3-oxo-C12-HSL and C8-HSL (Fig. 1c; see also Supplementary Fig. 2A, B). The $EC_{50}$ for the LasI product was 8.9 μM (Fig. 1d). Other AHLs (bacterially produced 3-oxo-C6-HSL from EsaI-transfected *E. coli* (Supplementary Table 1) and synthetic 3-oxo-C6-HSL) did not induce $Ca^{2+}$ responses from HEK293-cell-expressed Tas2r105 (Fig. 1c; see also Supplementary Fig. 2A, B). Tas2r105 was also activated by the bitter compounds denatonium benzoate (Den), a known activator of mouse nasal SCCs[23], and cycloheximide, produced by *Streptomyces griseus*, known to be bitter to mice[32,33] and a ligand for Tas2r105[34] (Supplementary Fig. 2C).

**Lack of gustducin increases alveolar bone loss (ABL).** A hallmark of periodontitis is ABL. Naturally occurring ABL develops slowly, being held in check by commensal oral microbiota[9]. ABL was assessed in wild-type (WT) and Gnat3−/− mice by measuring the distance between the cementoenamel junction (CEJ) of the second maxillary molar and the alveolar bone crest (ABC), which represents the alveolar bone level. Consistent with previous reports, the alveolar bone levels of WT mice declined between 8 and 16 weeks of age, although the loss was not statistically significant (Fig. 2a, b). Alveolar bone levels at 8 weeks of age were similar in WT and Gnat3−/− mice; in contrast, at 16 weeks of age Gnat3−/− mice exhibited significantly lower alveolar bone levels than did WT (with a longer distance from CEJ to ABC) (Fig. 2a, b). Micro-computed tomography (microCT) revealed that there were no obvious changes in alveolar bone density or trabecular number or thickness between WT and Gnat3−/− mice at 8 and 16 weeks (Fig. 2c–e), i.e., there were no baseline differences in these measures.

**Lack of gustducin alters the oral microbiome.** To characterize the commensal oral microbiota that lead to naturally occurring ABL, we performed 16S rDNA sequencing on oral swabs collected from WT and Gnat3−/− mice at weaning day, 8 weeks, and

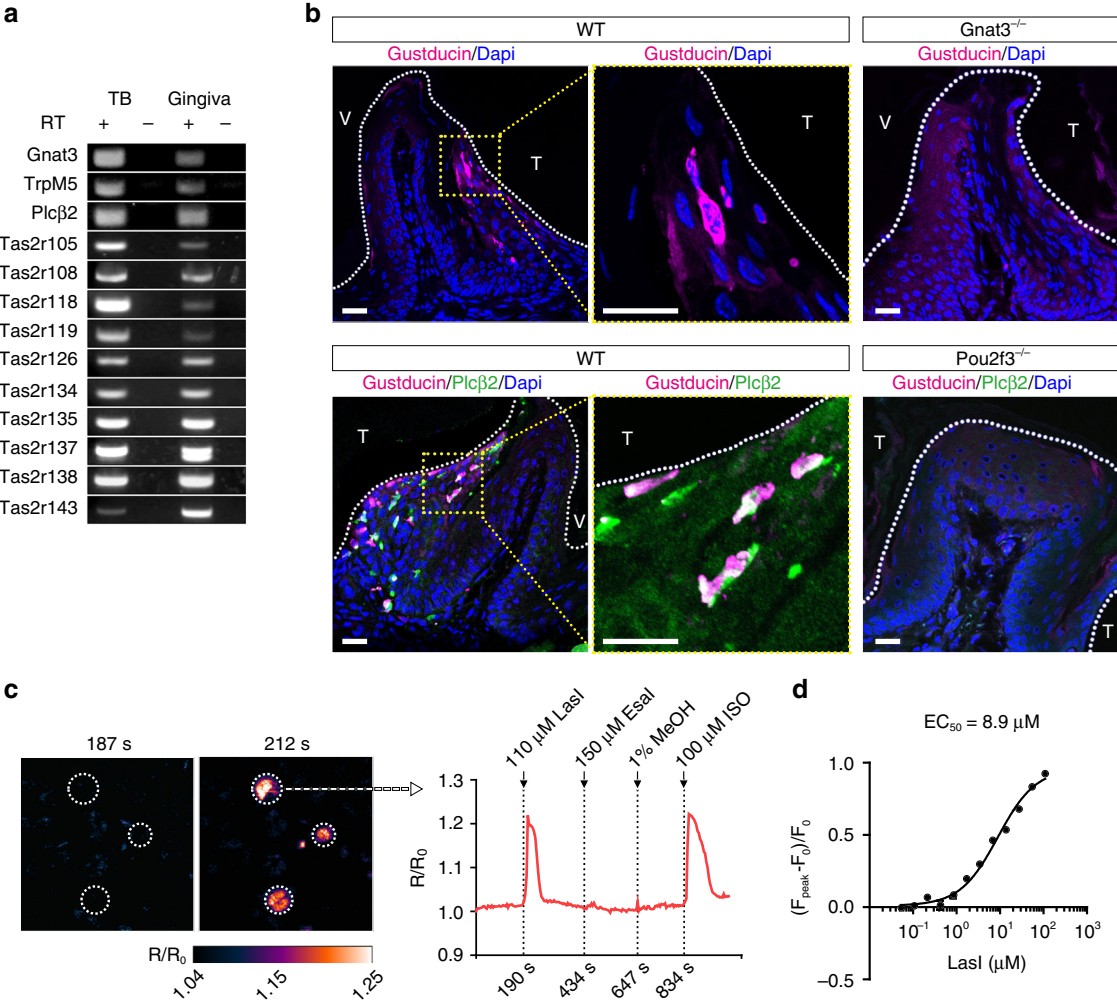

**Fig. 1** Solitary chemosensory cells in gingival epithelium. **a** Expression in gingiva of mRNAs for taste transduction elements examined by RT-PCR. TB taste buds, RT +/− with/without reverse transcription. Uncropped scans of the gel are provided in the Source Data file. **b** Expression in gingiva of gustducin (Gnat3) and phospholipase Cβ2 (Plcβ2) in wild-type (WT), Gnat3$^{-/-}$, and Pou2f3$^{-/-}$ mice. Nuclei stained by DAPI are blue. White dotted lines show tooth margins. T tooth facing side, V vestibular groove facing side. Yellow dotted lines indicate the fields with magnified views. Scale bars: 150 μm. **c** Responses of HEK293 cells transfected with Tas2r105 and chimeric reporter G protein (Gα16Gust44) to topically administered stimuli. The pseudocolor map represents calcium changes ($R/R_0$) measured by fluorescence intensity. LasI bacterially produced 3-oxo-C12-homoserine lactone (HSL), EsaI bacterially produced 3-oxo-C6-HSL, MeOH methanol vehicle control, ISO isoproterenol positive control. **d** Dose-dependent calcium responses to LasI of HEK293 cells transfected with Tas2r105 and Gα16Gust44. Source data are provided as a Source Data file

16 weeks. Principal component analysis (PCA) showed that the beta diversity of the oral microbiome changed over time in both types of mice (Fig. 2f, g; see Supplementary Table 2). At each time point, the commensal oral microbial communities from the two groups were distinguished by beta diversity (Fig. 2h–j; see also Supplementary Table 2). Analysis of prevalent bacterial taxa revealed genus-level differences between microbial communities from WT and Gnat3$^{-/-}$ mice (Supplementary Fig. 3). Of note, at all three time points Gnat3$^{-/-}$ mice had increased abundance of *Corynebacterium* and unranked *Cyanobacteria* and lower levels of *Muribacter* and *Porphyromonas* (Supplementary Fig. 3B). Taken together, these data indicate that the absence of α-gustducin in gSCCs has a marked effect on the oral microbial composition. Importantly, the differences in oral bacterial composition of WT vs. Gnat3$^{-/-}$ mice occurred before the loss of alveolar bone.

**Mice lacking SCC function develop more severe periodontitis**. To assess the impact of gSCCs and their taste signaling elements on periodontitis in mouse, we used molar ligation to induce periodontitis[15,35]. In all groups of mice, the placement of the ligature induced more extensive ABL at the ligatured site relative to the contralateral unligatured control site (Fig. 3a, b). Mice lacking SCC taste signaling molecules (i.e., Gnat3$^{-/-}$ mice) or lacking SCCs (i.e., Pou2f3$^{-/-}$ mice) developed more severe ligature-induced periodontitis with an increased level of ABL compared with WT mice (Fig. 3a, b). To assess differences in proinflammatory cytokines associated with ligature-induced periodontitis, we measured the cytokine mRNA levels in gingiva from ligatured and unligatured molars in WT and Gnat3$^{-/-}$ mice. While the expression of interleukin (IL)-1β, IL-6, and IL-17 and receptor activator of nuclear factor kappa-B ligand (RANKL) were enhanced in both WT and Gnat3$^{-/-}$ mice (Fig. 3c), the ligature-induced overexpression of these cytokines was markedly higher in Gnat3$^{-/-}$ than in WT mice (Fig. 3c), consistent with Gnat3$^{-/-}$ mice exhibiting much more severe ABL in ligature-induced periodontitis. Note that, based on measurements from the unligatured side, WT and Gnat3$^{-/-}$ mice had comparable absolute basal levels of IL-1β, IL-6, IL-17, and RANKL (Supplementary Fig. 4A).

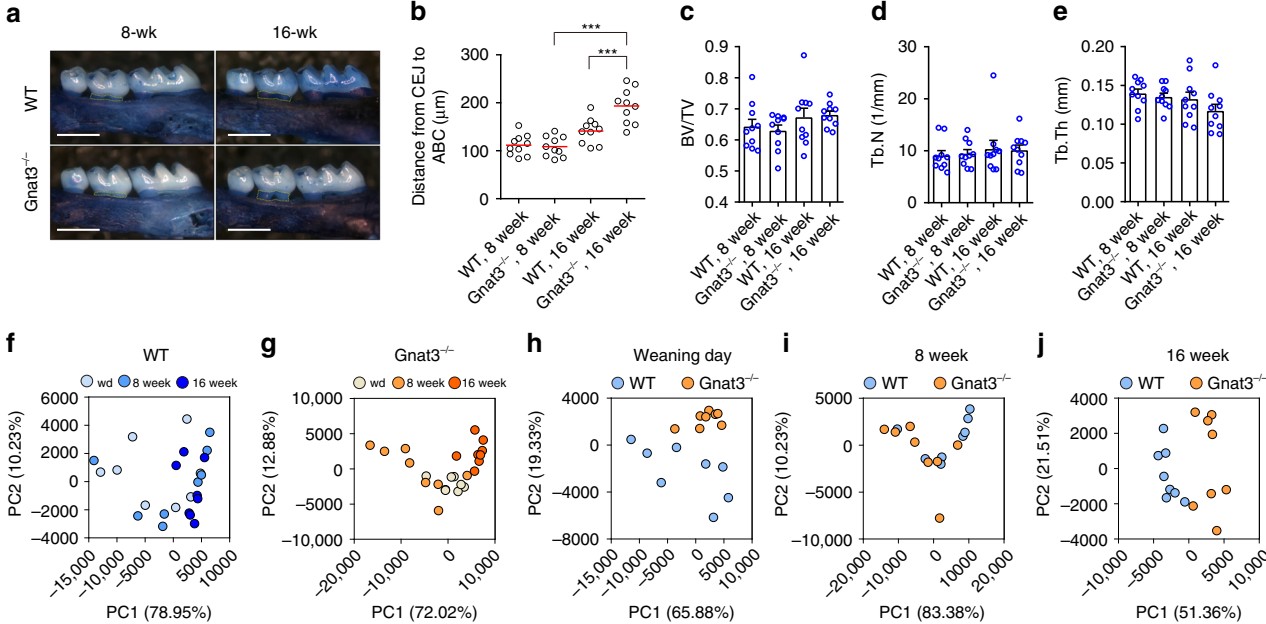

**Fig. 2** Accelerated naturally occurring alveolar bone loss and distinct commensal oral microbiota in Gnat3$^{-/-}$ mice. **a** Defleshed maxillae stained with methylene blue from wild-type (WT) and Gnat3$^{-/-}$ mice at 8 or 16 weeks of age. Yellow dotted line indicates the area between the cementoenamel junction (CEJ) of the second maxillary molar and the alveolar bone crest (ABC). Scale bars: 500 μm. **b** Quantitation of the distance from the CEJ of the second maxillary molar to the ABC. The result for each mouse is plotted; the red line indicates the mean ($n = 10$ mice). ***$p < 0.001$, one-way ANOVA test followed by Tukey's test. **c–e** MicroCT analysis of alveolar bone ($n = 10$ mice). BV/TV bone volume/tissue volume, Tb.N trabecular number, Tb.Th trabecular thickness. **f–j** Principal component analysis (PCA) of microbiota recovered from oral swabs collected from WT and Gnat3$^{-/-}$ mice at three time points: weaning day (wd) and 8 and 16 weeks of age. Each circle represents an individual oral swab sample ($n = 8$ mice), color coded by age (**f**, **g**) or genotype (**h–j**). Error bars in **c–e** represent the SEM. Source data are provided as a Source Data file

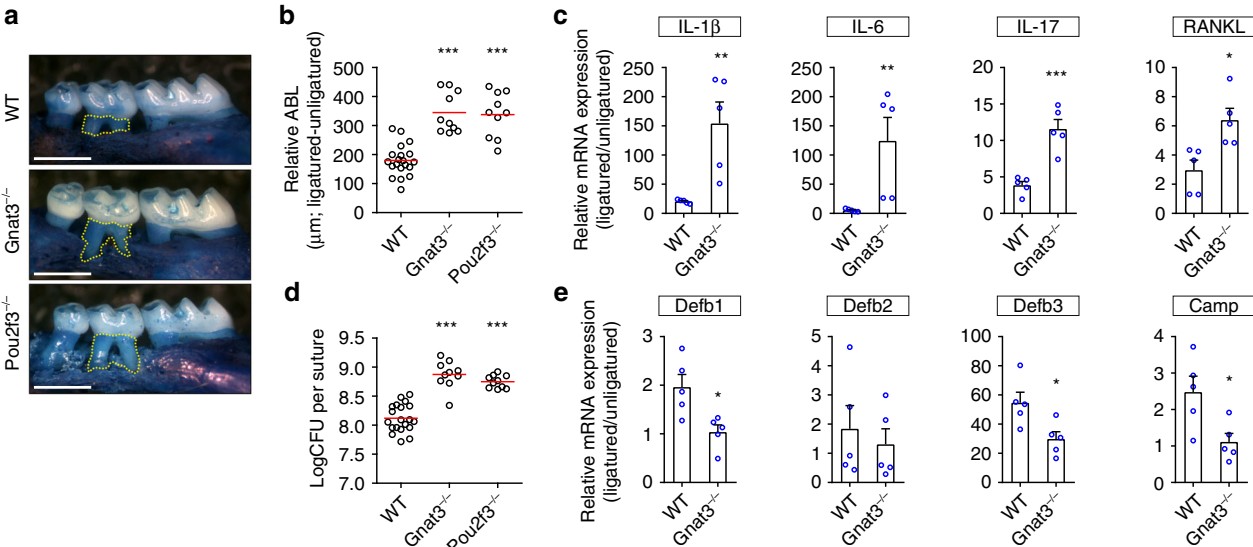

**Fig. 3** Ligature-induced periodontitis is more severe in mice lacking SCC signaling elements. **a** Ligatured maxillae from WT and knockout mice. Yellow dotted line indicates the area between the cementoenamel junction of the second maxillary molar and the alveolar bone crest. Scale bars: 500 μm. **b** Quantitation of relative alveolar bone loss (ABL) calculated by subtracting the ABL of the unligatured side from the ABL of the ligatured side. Results for each mouse are plotted; the red line indicates the mean ($n = 10$ for each knockout mouse and $n = 20$ WT mice). ***$p < 0.001$, one-way ANOVA test followed by Tukey's test. **c** Expression of pro-inflammatory cytokine mRNAs determined by qPCR. Results are normalized against β-actin mRNA expression and are represented as the fold change in transcript levels in ligatured sites relative to those of the corresponding contralateral unligatured sites, which are assigned a value of 1 ($n = 5$ independent experiments). IL-1β, -6, -17 interleukin-1β, -6, -17, respectively, RANKL receptor activator of nuclear factor kappa-B ligand. **$p < 0.01$, ***$p < 0.001$, Student's $t$ test. **d** qPCR quantitation of bacteria colonized on the ligatures recovered 1 week after placement. Result of each mouse is plotted; the red line indicates the mean ($n = 10$ knockout mice and $n = 30$ WT mice). CFU colony-forming unit. ***$p < 0.001$, one-way ANOVA test followed by Tukey's test. **e** Expression levels of antimicrobial peptides determined by qPCR ($n = 5$ independent experiments). Defb1–3 β-defensin 1–3, respectively, Camp cathelicidin antimicrobial peptide LL-37. *$p < 0.05$, Student's $t$ test. Error bars in **c** and **e** represent the SEM. Source data are provided as a Source Data file

Extensive ABL in ligature-induced periodontitis results from massive local bacterial accumulation in and around the ligatures[9,36]. We used quantitative real-time PCR (qPCR) to quantify the bacterial load on the ligatures, finding greatly increased bacterial colonization on the ligatures from Gnat3[−/−] mice (Fig. 3d). To determine whether overgrowth of bacteria on the ligatured molar was correlated with diminished secretion of AMPs in mice lacking Gnat3, we assessed mRNA expression of four AMPs in gingiva dissected from WT and Gnat3[−/−] mice. The expression of mRNAs for β-defensin-1 (Defb1), β-defensin-3 (Defb3), and LL-37 peptide from the cathelicidin antimicrobial peptide gene (Camp) was reduced by ~50% in Gnat3[−/−] vs. WT mice, while expression of β-defensin-2 (Defb2) mRNA was not altered (Fig. 3e). Measurements from the unligatured side showed that WT and Gnat3[−/−] mice had comparable absolute basal levels of Defb1, Defb2 and Camp, while basal Defb3 was reduced in Gnat3[−/−] mice (Supplementary Fig. 4B).

The pathogenicity of the oral microbiota relates to both the bacterial load and the microbial composition. To determine whether the absence of α-gustducin altered the composition of the commensal oral microbiota, we performed 16S rDNA sequencing of bacterial samples recovered from ligatures. We found that microbial communities on the ligatures from WT vs. Gnat3[−/−] mice differed in terms of beta diversity (Fig. 4a; see also Supplementary Table 2) and microbial network (Fig. 4b). In addition, the diversity of the microbiota within individual samples (alpha diversity) was significantly lower in the ligatures from the Gnat3[−/−] mice (Supplementary Fig. 5). Furthermore, genera *Pasteurella*, *Streptococcus*, and *Gemella* were enriched in the ligature microbiota from Gnat3[−/−] vs. WT mice, while the proportions of *Porphyromonas*, *Enterococcus*, and *Proteus* were decreased (Fig. 4c, d). The enrichment of *Pasteurella* may be significant: a pathogen (NI1060) belonging to this genus has been associated with ligature-induced periodontitis[15]. Therefore, we used qPCR to quantify the abundance of this pathogen in the bacterial samples recovered from ligatures, finding that the percentage of NI1060 was significantly higher in ligatures from Gnat3[−/−] than in those from WT mice (Fig. 4e). To determine whether differences in oral microbiota between Gnat3[−/−] and WT mice were due to the genetic loss of Gnat3 and not from adventitious or natural variation, we cohoused Gnat3[−/−] mice and WT littermates from birth until 8 weeks of age, placed molar ligatures, and then housed ligatured Gnat3[−/−] and WT mice in separate cages to allow development of bacteria around the ligatured molars. Consistent with results of our previous ligature experiments, after cohousing the Gnat3[−/−] mice developed more severe ligature-induced periodontitis than did WT mice, with an increased level of ABL and decreased bone volume per tissue volume (BV/TV; see Supplementary Fig. 6A–C). Moreover, cohoused Gnat3[−/−] and WT littermates developed distinct microbial communities on their ligatures (based on analysis of β diversity; see Supplementary Fig. 6D), indicating that it was the genetic deficiency of Gnat3, and not naturally different microbiota, that differentially affected microbiota diversity. The Shannon index of the microbiome from ligatures of cohoused Gnat3[−/−] mice were also lower than that of the microbiome from ligatures of WT mice (see Supplementary Fig. 6E). Although differences in the Shannon indices did not reach statistical significance, the α diversity of the oral microbiome from Gnat3[−/−] mice were reduced compared to those of WT (see Supplementary Fig. 6E). Altogether, these results indicate that the microbiota colonized on ligatures in Gnat3[−/−] mice compared to those of WT mice are characterized by increased bacterial load, diminished diversity, and increased levels of pathogens.

**Activation of gingival SCCs protects against periodontitis**. To examine the effects of repeated stimulation of gSCCs on periodontitis and gingival AMPs, we topically applied the bitter compound Den (1 mM), a well-established activator of SCCs and a ligand for Tas2r105 (Supplementary Fig. 2C), to the mouse gingiva twice daily (Fig. 5a). After treatment with Den, the gingival expression of β-defensin-3 (Defb3) in WT mice was enhanced more than twofold compared with controls treated with phosphate-buffered saline (PBS; Fig. 5b). In contrast, treatment with Den in Gnat3[−/−] mice had no effect on the expression level of Defb3 (or the other AMPs tested) (Fig. 5b).

To characterize the protective function of gSCCs and determine whether their secretion of Defb3 after treatment with Den could repress bacterial colonization and thus protect against periodontitis, we performed molar ligation and then treated the mice with Den or PBS twice daily for 6 days (Fig. 5c). Den treatment of WT mice significantly decreased ABL compared with PBS-treated controls (Fig. 5d, e). In contrast, treatment of Den had no effect on ABL in Gnat3[−/−] mice (Fig. 5d, e). Consistent with the ABL results, the ligature bacterial load among the four groups was lowest in WT mice treated with Den but was unaffected in Gnat3[−/−] mice by treatment with either Den or PBS (Fig. 5f), demonstrating the necessity of functional gSCCs for denatonium-induced protection against pathogenic bacteria.

## Discussion

SCCs, first characterized in the mouse nasal cavity[24], are emerging as key sentinels that respond to microbial metabolites to initiate host innate immunity[17,20,21,23]. A defining feature of SCCs is the expression of taste receptors along with associated downstream taste signaling molecules, such as α-gustducin, Plcβ2, and TrpM5. Here we show that SCCs are present in the gingiva and that these gSCCs express 10 Tas2r bitter receptors along with α-gustducin, Plcβ2, and TrpM5. Upon activation, most SCCs, including cholinergic mouse nasal SCCs, can release the neurotransmitter acetylcholine to trigger downstream effects[22]. However, SCCs found in urethra comprised at least two distinct populations, one ChAT positive and one ChAT negative[17,37]. The gSCCs do not express ChAT, suggesting they may release neurotransmitters other than acetylcholine or bioactive peptides as downstream signals.

In mouse nasal SCCs, bitter receptors detect bacterially derived substances such as AHLs, while their downstream taste transduction components couple this detection to an activating signaling cascade[23]. In this study, we confirmed that bacterially produced or synthetic AHLs 3-oxo-C12-AHL and C8-HSL activated Tas2r105, 1 of the 10 Tas2rs expressed in gSCCs. Consistent with our data, a recent in vitro study[34] found that Tas2r105 was the only mouse Tas2r receptor that responded to four tested AHLs (3-oxo-C6-HSL, 3-oxo-C8-HSL, C4-HSL, and C6-HSL) and that Tas2r105 could be activated by Den and cycloheximide as well. Although mouse nasal SCCs could be activated by Den and AHLs[23], previous in situ hybridization studies did not detect the expression of Tas2r105 mRNA in mouse nasal SCCs[24]. Urethral SCCs were shown by RT-PCR to express Tas2r108 but not Tas2r105 and Tas2r119[17]. Consistent with this result, urethral SCCs were not activated by the Tas2r105 agonist cycloheximide. It seems likely that urethral SCCs are specialized to detect irritants and bacterial metabolites via Tas2rs other than Tas2r105.

A striking feature of SCCs is the co-expression of Tas1r sweet/umami taste receptors along with Tas2r bitter taste receptor[17,20,21]; this contrasts with taste receptor cells that typically express either Tas1rs or Tas2rs but not both[32]. In the gastrointestinal tract, tuft cells could induce type 2 immunity against

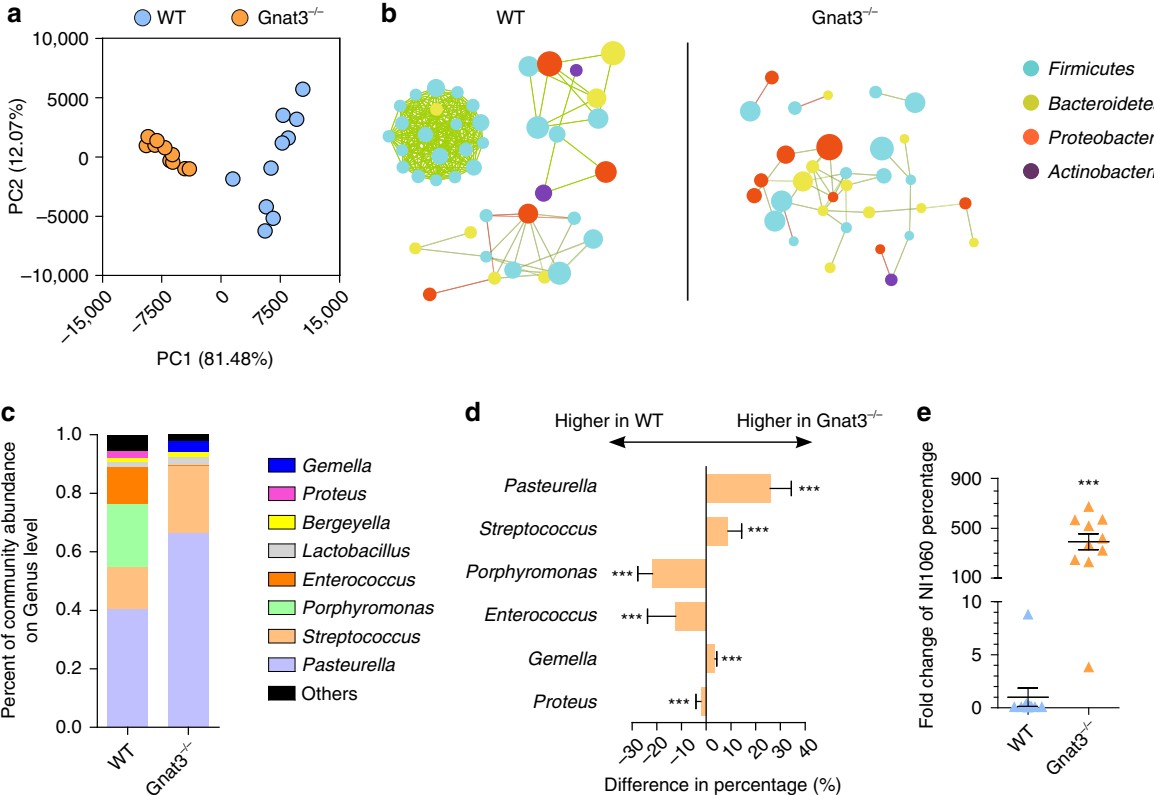

**Fig. 4 Gnat3$^{-/-}$ mice develop distinct oral microbiota. a** Principal component analysis (PCA) of microbiota recovered from ligatures around molars of WT and Gnat3$^{-/-}$ mice ($n = 10$ mice). Each circle represents an individual ligature sample, colored by genotype. **b** Operational taxonomic unit (OTU)-level correlation network analysis of ligature microbiota from WT and Gnat3$^{-/-}$ mice ($n = 10$ mice). The most abundant 50 OTUs from each genotype were analyzed. Each circle represents one OTU, colored by phylum. The size of a circle indicates the abundance of the OTU. Green lines indicate positive correlations between two OTUs; red lines indicate negative correlations. **c** Average genus-level composition of ligature microbiota from WT and Gnat3$^{-/-}$ mice ($n = 10$ mice). **d** Prevalent genus with significant difference in abundance between WT and Gnat3$^{-/-}$ mice ($n = 10$ mice, means ± 95% confidence interval). ***$p < 0.01$, Wilcoxon rank-sum test. **e** qPCR quantification of NI1060, a ligature-induced periodontitis-related pathogen. Result of each ligature sample is plotted, with black line indicating means ± SEM ($n = 10$ mice). ***$p < 0.01$, Wilcoxon rank-sum test. Source data are provided as a Source Data file

parasite infection in a taste-signaling-dependent way[18]. It has been proposed that tuft cells might use other metabolite-sensing G protein-coupled receptors to monitor the microbial component in the gastrointestinal tract[38]. Indeed, tuft cells express short-chain fatty acid receptors Gpr41 and Gpr43, along with α-gustducin, providing a mechanism for bacterial metabolites to trigger host responses[25]. Intriguingly, Gpr41 and Gpr43 were also expressed in mouse gingiva, potentially in gSCCs. It would be interesting to determine whether gSCCs do indeed use these short-chain fatty acid receptors to mediate responses to bacteria.

The current study adds to the breadth of tissues that contain SCCs and indicates that the canonical taste transduction cascade in gSCCs is involved in the regulation of oral microbiota. A growing body of evidence suggests that periodontitis is caused by interactions within dysbiotic polymicrobial communities[1,4–8]. Interestingly, well-documented periodontal pathogens such as *Porphyromonas gingivalis* could not induce ABL in germ-free mice without commensal oral bacteria[9]. In the absence of taste signaling in gSCCs (i.e., Gnat3$^{-/-}$ mice), the commensal oral microbial composition was altered and naturally occurring ABL was enhanced. However, since the murine oral microbiota has not been thoroughly investigated, it is uncertain whether the genus-level microbial alteration observed here is either a correlate or a cause of the increased ABL in Gnat3$^{-/-}$ mice. It is important to note that the changes in commensal bacterial composition took place at an early age (weaning day, ~3 weeks old), suggesting the microbial alteration could be an initial factor rather than an

outcome of accelerated naturally occurring ABL. Nevertheless, more in-depth studies are required to delineate the cause–effect relationship. Another possibility is that lack of taste signaling components in gSCCs led to an overgrowth of oral bacteria colonized on molar ligatures, making the mice more vulnerable to ligature-induced periodontitis. In this regard, it is important to note that the microbiota recovered from ligatures of Gnat3$^{-/-}$ mice also showed a reduced alpha diversity, which is generally a sign of less stable and more pathogenic microbial communities[5,39]. Consistent with this, we found that a previously identified ligature-induced periodontitis-related commensal NI1060[15] was enriched in Gnat3$^{-/-}$ mice. All these shifts in ligature microbiota suggest increased pathogenicity in the absence of functional gSCCs. Oral microbiota transplantation would be a useful experimental approach to determine whether WT mice transplanted with the microbiota from Gnat3$^{-/-}$ mice can restore normal microbiome equilibrium by eliminating the excess pathogenic bacteria through the activity of functional gSCCs.

Under periodontal homeostasis in mice, oral commensal microbiota induce only modest ABL, with controlled inflammation[9]. The placement of ligature causes tissue damage that disturbs the balanced periodontal ecosystem, leading to a dysbiotic microbiome and extensive ABL from resultant excessive inflammation[15,36]. In this study, mice devoid of taste signaling molecules in their SCCs or lacking SCCs exhibited increased susceptibility to and severity of ligature-induced periodontitis. The defective immune surveillance allowed the overgrowth of

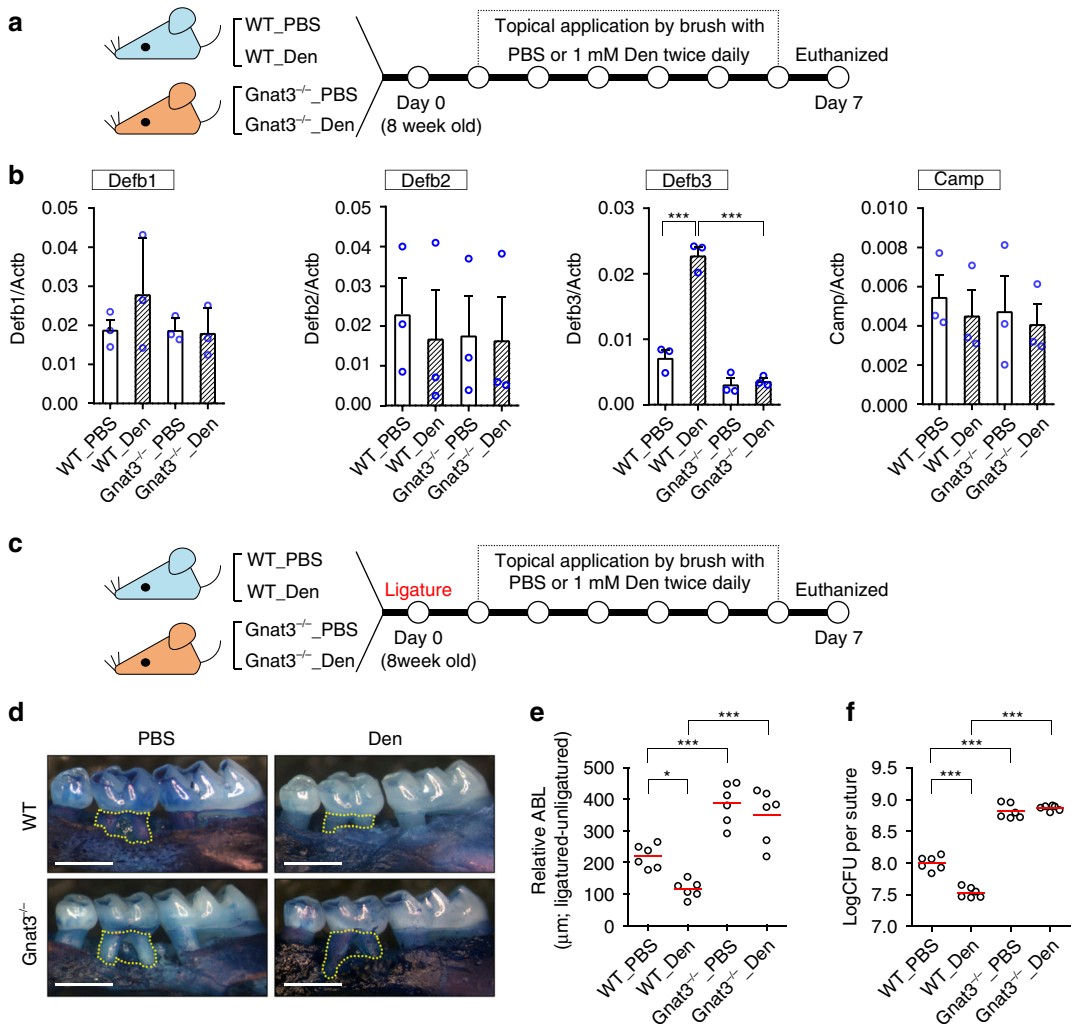

**Fig. 5** Activation of gSCCs stimulated expression of beta-defensin and alleviated periodontitis in wild-type mice. **a** Molars of WT and Gnat3$^{-/-}$ mice were topically treated with 1 mM denatonium benzoate (Den) or PBS for 6 days and then mice euthanized at day 7. Gingival tissues were collected for RNA isolation and qRT-PCR. **b** Gingival expression of antimicrobial peptide mRNAs determined by qRT-PCR. Results are normalized against β-actin mRNA (Actb). Data are means ± SEM ($n = 3$ mice). Defb1–3 β-defensin 1–3, respectively, Camp LL-37. ***$p < 0.001$, one-way ANOVA test followed by Tukey's test. **c** At day 0, silk ligatures were placed around the second maxillary left molars of WT and Gnat3$^{-/-}$ mice. The mice were treated with 1 mM Den or PBS from day 1 to day 6 and sacrificed on day 7. **d** Maxillae from ligatured WT and Gnat3$^{-/-}$ mice treated with PBS or Den. Yellow dotted line indicates the area between the cementoenamel junction of the second maxillary molar to the alveolar bone crest. Scale bars: 500 μm. **e** Quantitation of relative ABL calculated by subtracting the ABL of the unligatured side from the ABL of the ligatured side. Result of each mouse is plotted; the red line indicates the mean ($n = 6$ mice). *$p < 0.05$; ***$p < 0.001$; one-way ANOVA test followed by Tukey's test. **f** Quantitation by qPCR of the bacteria colonized on the ligatures recovered 1 week after placement. Result of each mouse is plotted, with the red line indicating the mean ($n = 6$ mice). ***$p < 0.001$, one-way ANOVA test followed by Tukey's test. Source data are provided as a Source Data file

normally benign commensal bacteria, especially in the damaged tissue sites. The increased microbial burden in turn amplified the tissue inflammation and damage, fostering nutrient leakage from damaged tissue and favoring further bacterial growth. In addition to changes in bacterial load, the composition of the microbiota was altered as well, with increased NI1060. Our results indicate that the overgrowth of commensal bacteria might be a result of depressed gingival AMP secretion in knockout (KO) mice. In contrast, topical treatment of WT mice with the SCC activator Den stimulated the expression of gingival AMP Defb3, consistent with previous studies showing that bitter compounds affect human airway immunity[20,31]. Intriguingly, human Defb3 has been reported to be positively related with periodontal health[40]. Moreover, in the ligature-induced periodontitis model, such treatment alleviated the ABL level in WT mice. In this regard, the activation of gSCCs in WT mice limited the overgrowth of

commensal bacteria on ligatures probably via Defb3 secretion, thus preventing further deterioration in microbial dysbiosis. We hypothesize that the deficiency in taste signaling in gSCCs causes aberrant or insufficient AMP secretion, leading to overgrowth of oral bacteria and formation of a dysbiotic microbiota responsible for the increased susceptibility to periodontitis (Fig. 6).

In human upper airway, TAS2R38 is involved in detecting AHLs and plays a role in preventing Gram-negative bacterial infection, such as chronic rhinosinusitis[30,41]. Intriguingly, TAS2R38 has a uniquely high density of naturally occurring genetic variants[42]. AHLs and canonical bitter ligands phenylthiocarbamide and 6-n-propylthiouracil can activate the functional TAS2R38 but not the nonfunctional form. TAS2R38 has been found in various tissues/cell types[43–45], including cultured human gingival epithelial cells[43]. Furthermore, clinical studies showed that functional TAS2R38 was correlated with

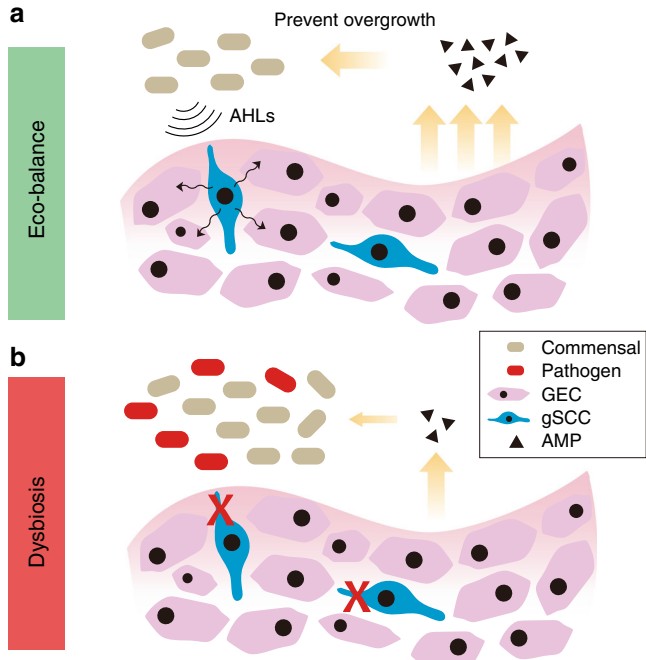

**Fig. 6** Potential roles of gingival SCCs in oral microbiome regulation. **a** Taste-like gSCCs in mouse gingival tissue may detect bacterial signals (e.g., acylated homoserine lactones (AHLs)) via bitter taste receptors (Tas2rs), thus triggering host innate immune responses (e.g., antimicrobial peptide (AMP) secretion) to prevent the overgrowth of oral bacteria and also regulate the microbial composition. **b** The dysfunction of gSCCs may cause insufficient AMP secretion, rendering a dysbiotic microbiota characterized with increased bacterial load, diminished diversity, and increased levels of pathogens (e.g. NI1060). GEC gingival epithelial cells

protection against caries[46,47]. More recently, a study proposed that the activation of functional TAS2R38 by cariogenic *Streptococcus mutans* could trigger antimicrobial activity in cultured human gingival epithelial cells and thus may have a protective effect against caries[43]. From our study in mouse, it is plausible that TAS2R38 and downstream signaling molecules are also expressed in human gSCCs, presumably playing a similar role in regulating oral microbiota and protecting against periodontitis.

In summary, mouse gSCCs likely respond to bacterial signals via Tas2rs and downstream taste signaling components to trigger host innate immune responses, such as secretion of AMPs, to prevent overgrowth of oral bacteria and regulate oral microbial composition (Fig. 6). Further studies from mouse models and clinical cohorts are needed to delineate the role of gSCCs in pathogenesis of periodontitis. Importantly, since polymorphisms in taste receptors, particularly the loss-of-function genotype of the bitter taste receptor TAS2R38, are commonly detected in populations, the dysfunction of taste-receptor-mediated innate immune responses could be utilized for dental chair-side screening of non-tasters as potential susceptible individuals of oral infectious diseases. This may lead to a new paradigm of personalized dental treatments against oral infectious diseases.

## Methods

**Contact for reagent and resource sharing**. Further information and requests for resources and reagents should be directed to and will be fulfilled by the Lead Contact, Robert F. Margolskee (rmargolskee@monell.org).

**Mouse models**. The design and production of α-gustducin-null (Gnat3$^{-/-}$) mice[48], Pou2f3-null (Pou2f3$^{-/-}$) mice[49], mice expressing ChAT-driven GFP (ChAT-GFP)[50], and TrpM5-driven GFP (TrpM5-GFP)[51] were as described previously (Supplementary Table 1). The genetic background of all mice was C57BL/

6J, except for ChAT-GFP mice, which were generated from FVB/N mice. All mice were housed in the same environment under specific pathogen-free conditions with a 12-h light/dark cycle and ad libitum access to food and water. Mice were sacrificed by $CO_2$ euthanasia followed by cervical dislocation. All animal experiments were performed in accordance with the National Institutes of Health guidelines for the care and use of animals in research and approved by the Institutional Animal Care and Use Committee at Monell Chemical Senses Center (ACC# 1184).

For RT-PCR, 8 WT mice (C57BL/6J) aged ~24 weeks of both sexes were used for collection of taste bud and gingival tissues. For histologic studies, 8 WT, Gnat3$^{-/-}$, TrpM5-GFP, ChAT-GFP, and Pou2f3$^{-/-}$ mice aged ~16 weeks of both sexes were used. For observation of naturally occurring ABL and microCT analyses, 10 mice of both sexes for each of the 4 groups (WT and Gnat3$^{-/-}$ of 8 or 16 weeks of age) were used. For commensal oral microbiota analysis, oral swabs were collected from 4 males and 4 females in both the WT and Gnat3$^{-/-}$ groups, at 3 time points (weaning day, 8 weeks, and 16 weeks).

For ligature-induced periodontitis, all mice used were approximately aged 8 weeks. Initial experiments used five WT and five KO mice and were repeated twice. In total, 20 WT mice and 10 of each type of KO mice were included. WT and Gnat3$^{-/-}$ mice used were of both sexes while Pou2f3$^{-/-}$ mice were all male. To investigate the effect of cohousing on ligature-induced periodontitis, we cohoused 3 WT mice and 3 Gnat3$^{-/-}$ mice from birth until 8 weeks of age, placed molar ligatures, and then housed separately by genotype.

For treatment with bitter Den, 6 mice aged ~8 weeks of both sexes were included in each of the four groups (i.e., WT or Gnat3$^{-/-}$ mice stimulated with Den or PBS). WT and Gnat3$^{-/-}$ mice were randomly assigned to the Den treatment group or the PBS treatment group.

**Cell culture**. For functional tests of mouse bitter taste receptors, human embryonic kidney 293 (HEK293) PEAKrapid cells were obtained from ATCC and cultured in Opti-MEM (Thermo Fisher Scientific, Waltham, MA) supplemented with 4% fetal bovine serum, at 37 °C under aerobic condition (5% $CO_2$). For transfections, cells were seeded in 96-well plates at a density of 60,000 per well.

**RNA extraction and RT-PCR**. Gingival tissues were peeled off from mouse maxilla under a dissecting microscope and cut into pieces. For ligature-induced periodontitis assays, only the gingival tissues around the maxillary second molar were collected. For preparation of taste bud tissues, tongues were excised from WT mice. Lingual epithelia were removed as previously described[52]. An enzyme mixture consisting of dispase II (2 mg/ml; Roche, Mannheim, Germany; cat. no. 04942078001) and collagenase A (1 mg/ml; Roche; cat. no. 10103578001) in $Ca^{2+}$-free Tyrode's solution (145 mM NaCl, 5 mM KCl, 10 mM HEPES, 5 mM NaHCO$_3$, 10 mM pyruvate, 10 mM glucose) was injected below the lingual epithelium. The tongue was then incubated for 15 min at 37 °C. Lingual epithelium was peeled off and fixed in a Sylgard-coated petri dish, and epithelium containing circumvallate taste papillae was dissected from the surrounding tissue and cut into small pieces. All surgical instruments were treated with RNaseZap (Thermo Fisher Scientific). Tissues collected were all transferred into lysate (included in the PureLink™ RNA Mini Kit; Thermo Fisher Scientific; cat. no. 12183025) with 1% β-mercaptoethanol added, followed by immediate RNA extraction or storage at −80 °C.

RNA was isolated and purified according to the manufacturer's instructions for the PureLink™ RNA Mini Kit. The RNA samples were further treated with RQ1 RNase-Free DNase (Promega, Madison, WI; cat. no. M6101) to remove genomic DNA contamination and then reverse transcribed using SuperScript™ VILO™ Master Mix (Thermo Fisher Scientific; cat. no. 11755050) with random hexamer primers. Aliquots without RT were prepared as negative controls. PCR was performed using Platinum™ *Taq* DNA Polymerase (Thermo Fisher Scientific; cat. no. 10966026). Primers for all 35 mouse Tas2rs, Gnat3, TrpM5, Plcβ2, Gpr41, and Gpr43 are listed in Supplementary Table 3.

**Histology**. Maxillae were dissected from WT, Gnat3$^{-/-}$, Pou2f3$^{-/-}$, ChAT-GFP, and TrpM5-GFP mice and fixed in fresh 4% paraformaldehyde in 1 × PBS for 2 days at 4 °C. Fixed tissues were immersed in 14% EDTA in 1 × PBS for 1 month at room temperature for decalcification; the solution was refreshed every other day. Before embedding in Tissue-Tek O.C.T. mounting media (Sakura, Netherlands), decalcified maxillae were cryoprotected in 20% sucrose in 1 × PBS overnight at 4 °C. Tissue sections (6–8 μm) were obtained with a cryostat microtome (Leica, Wetzlar, Germany). Sections were dried for 45 min at 42 °C and used for immunofluorescence procedures immediately or stored at −80 °C.

For immunofluorescence staining, sections were washed three times for 5 min with 1 × PBS (frozen sections were dried for 45 min under 42 °C first) and blocked for 45 min with SuperBlock blocking buffer (Thermo Fisher, cat. no. 37515) supplemented with 0.3% Triton X-100 and 2% donkey serum, at room temperature. Sections were then incubated at 4 °C overnight with the desired primary antibodies for gustducin (1:100; Santa Cruz, Dallas, TX; cat. no. SC-395), Plcβ2 (1:100; Santa Cruz; cat. no. SC-31759), or GFP (1:500; Millipore, Germany; cat. no. AB16901). After washing 3× for 5 min with 1 × PBS, sections were incubated for 1 h at room temperature with species-specific secondary antibodies conjugated with different fluorophores (1:500; listed in Supplementary Table 1). 6-

Diamidino-2-phenylindole (1:1000; Molecular Probes) in deionized water was used to visualize the nuclei.

Fluorescent images were captured with the TCS SP2 Spectral Confocal Microscope (Leica) equipped with UV, Ar, GeNe, and HeNe lasers. Scanware software (Leica) was used to acquire $z$-series stacks captured at a step size of 2–3 μm. Acquisition parameters (gain, offset, and pinhole settings) were held constant for experiments with WT tissue and KO tissues. Fluorescence images within a figure were adjusted for brightness and contrast for background standardization by Image J (https://imagej.nih.gov/ij/).

**Functional assays of mouse bitter taste receptors**. The coding regions of the 10 mouse Tas2rs expressed in gingival tissue were amplified from cDNA from WT mouse taste bud tissue. To improve plasma membrane targeting, DNA encoding the first 45 amino acids of the rat somatostatin receptor subtype 3 was added to the 5′ end of each Tas2r construct by overlapping PCR. The resulting products were subcloned into expression plasmid pcDNA™3.1/Zeo$^{(+)}$ (Thermo Fisher Scientific; cat. no. V86020). Each Tas2r (100 ng/well) construct and the chimeric G protein Gα16gust44 construct (100 ng/well; provided by P.J.) were transiently transfected into ~90% confluent HEK293 cells in 96-well plates using Lipofectamine 2000 (0.5 μl/well; Thermo Fisher Scientific; cat. no. 11668019) according to previous procedures[34,53].

Calcium mobilization was monitored as previously described[34,53]. Transfected cells were washed once with Dulbecco's PBS (DPBS) with calcium and magnesium (Thermo Fisher Scientific; cat. no. 14040117) and loaded with Fluo-4 (2.5 μM in dimethyl sulfoxide (DMSO); Thermo Fisher Scientific; cat. no. F14201) for 1 h under room temperature in the presence of Pluronic F-127 (Thermo Fisher Scientific; cat. no. P3000MP). After three washes with DPBS, cells were incubated in the dark for another 30 min for complete de-esterification of the dye and then assayed for their responses to test compounds using a FlexStation III (Molecular Devices, Sunnyvale, CA). Relative fluorescence units (excitation at 488 nm, emission at 525 nm, and cutoff at 515 nm) were read every 2 s after addition of a volume of DPBS equal to that in the well that was supplemented with a twofold concentration of the test compound. For bar graph plots, changes in fluorescence were quantified as peak fluorescence minus the baseline level ($F_{peak} - F_0$; $F_0$ was the average of the last 20 readings of each well) and were expressed as ($F_{peak} - F_0$)/$F_0$. In addition, curves of fluorescence intensity over time, normalized by baseline ($F/F_0$), were drawn. The plotting of dose-dependent curves and calculation of EC$_{50}$ values were performed by the Prism software (GraphPad Software Inc., La Jolla, CA).

For calcium imaging assays, the calcium indicator dye Fura-2 (Thermo Fisher Scientific; cat. no. F14185) ratio imaging was used to measure intracellular Ca$^{2+}$ levels[54]. The transfected cells were split into 96-well plates at 1:20 dilution rate and were cultured overnight. The next day, the cells were incubated with Fura-2 (2.5 μM in DMSO) for 50 min at room temperature in the presence of Pluronic F-127, washed twice with DPBS, and then incubated an additional 30 min. Cells were examined under an IX2-UCB fluorescent microscope (Olympus, Japan). The excitation wavelength was alternated between 340 and 380 nm using a filter wheel incorporated into a xenon lamp system (Lambda XL; Sutter Instruments, Novato, CA). Pairs of images at the two wavelengths were acquired every 5 s. At different time points, certain stimuli were added into the well for 30 s and then flushed away by adding DPBS. The ratio of fluorescence intensity at excitation wavelengths of 340 and 380 nm was calculated and normalized with baseline ($R/R_0$; $R_0$ was the average ratio of first 10 pairs of images), and pseudocolor maps representing the $R/R_0$ were generated with Image J.

The commercial compounds used in the functional test of bitter taste receptors were all purchased from Sigma-Aldrich (St. Louis, MO), including 3-oxo-C12-HSL, 3-oxo-C6-HSL, C8-HSL, cycloheximide, Den, and isoproterenol. The bacterially produced AHLs LasI and EsaI were obtained from Dr. Mair Churchill (University of Colorado Denver) and were as previously described[23]. Two sets of negative controls were done: (a) HEK293 cells transfected with empty vector plus Gα16gust44, and (b) HEK293 cells transfected with Tas2r clone plus Gα16gust44 stimulated with solvent (e.g., DPBS, methanol, or DMSO). For more details, see Supplementary Table 1.

**Measurement of alveolar bone level**. Maxillae were dissected from WT and Gnat3$^{-/-}$ mice (8 or 16 weeks old) and from WT, Gnat3$^{-/-}$, and Pou2f3$^{-/-}$ mice in the ligature-induced periodontitis model. Collected maxillae were defleshed by immersing in 5% sodium hypochlorite for 15 min, stained with methylene blue for 1 min, and then washed with water. Stained maxillae were examined and photographed under a stereomicroscope equipped with a digital camera. The distance from the CEJ of the second maxillary molar to the ABC was measured at three different sites in the buccal side (distobuccal cusp, buccal groove, and mesiobuccal cusp) with Image-Pro Plus 6.0 (Media Cybernetics, Silver Spring, MD). The average value of these three sites represented the alveolar bone level of the maxillae. The bone level changes between 8- and 16-week-old mice or between the ligatured and unligatured side in one mouse are referred to, respectively, as naturally occurring ABL or relative ABL.

**MicroCT analyses of alveolar bone**. Maxillae were dissected from WT and Gnat3$^{-/-}$ mice (8 or 16 weeks old). Defleshed maxillae were scanned using microCT (μCT 50, SCANCO, Switzerland). The X-ray beam was set at 70 kVp and 200 μA. All samples were scanned in the sagittal position at a voxel resolution of 12 μm. Histomorphometric analysis of samples at the region of interest (see below) was performed by the CT-Analyser 1.13 software (Bruker, Belgium); parameters measured included BV/TV, trabecular number, trabecular thickness, and trabecular separation. The region of interest was defined as a rectangular region immediately below the top of bone septum between mesial and distal roots of the maxillary second molar.

**Microbial genomic DNA extraction**. Oral swabs were collected from the same mouse at weaning day and at 8 and 16 weeks of age from the WT and Gnat3$^{-/-}$ groups ($n = 8$ for each group). Oral bacterial samples were obtained by PurFlock™ Ultra Sterile Flocked Swabs (Puritan, Guilford, ME; cat. no. 253318U BT) of the teeth and gingival surfaces. In the ligature-induced periodontitis assay, ligatures were collected after the mice were euthanized ($n = 10$ in Fig. 3d; $n = 6$ in Fig. 5f). The microbial genomic DNAs were isolated using the PureLink™ Microbiome DNA Purification Kit (Thermo Fisher Scientific; cat. no. A29790) according to the manufacturer's instructions. The DNA quality was evaluated with a NanoDrop 2000 spectrophotometer (Thermo Fisher Scientific), and final concentration was quantified via the Pico-Green Kit (Thermo Fisher Scientific; cat. no. P11496). The DNA samples were used immediately for bacterial quantification by qPCR or stored at −80 °C.

**Microbiome analysis**. The library preparation and sequencing data analyses were performed as previously described[55]. 27F (5′-AGAGTTTGATCCTGGCTCAG-3′) and 533R (5′- GTGCCAGCAGCCGCGGTAA-3′) primers were used to amplify the V1–V3 region of 16S rDNA. A unique 12-mer tag for each DNA sample was added to the 5′-end of both primers to pool multiple samples for one run. PCR products were visualized on 3% agarose gels, gel purified, quantified with the Pico-Green Kit, pooled to an equal molar concentration, and assessed using Agilent BioAnalyzer 2100 (Thermo Fisher Scientific). At Majorbio Co. (Shanghai, China), barcoded 16S rDNA amplicon sequencing was performed through Illumina MiSeq technology, where on average 498-bp-long reads were produced. Sequences were trimmed using Trimmomatic[56] based on quality scores of 20, and pair-end reads were merged into longer reads by FLASH[57]. Unqualified sequences were removed if they were too short or contained ambiguous residues. Operational taxonomic units (OTUs) were clustered using Usearch version 7.1 (http://drive5.com/uparse/) at the 97% similarity level, and final OTUs were generated based on the clustering results. The sequencing raw data have been deposited in Sequence Read Archive (http://www.ncbi.nlm.nih.gov/Traces/sra; accession nos. SRP126006 and SRP215896).

The pre-processed sequencing data was further analyzed with the following statistical methods. (1) PCA was used to compare the beta diversity within groups. Two non-parametric analyses for multivariate data, analysis of similarities (ANOSIM) and multivariate analysis of variance (Adonis) using distance matrices, were used to examine the community difference within groups. (2) Taxonomic annotations were assigned to each OTU's representative sequence by blasting with the oral "CORE" reference database. The relative abundances of bacterial taxa at genus levels were analyzed. (3) Alpha diversity analysis was based on Shannon, Simpson, Ace, and Chao indices. (4) OTU-level microbial correlation network analysis was also performed on the most abundant 50 OTUs in the WT and Gnat3$^{-/-}$ groups. All analyses were performed with I-Sanger online tools (http://www.i-sanger.com/).

**Ligature-induced periodontitis**. Mice (8 weeks old) were anesthetized with an intraperitoneal injection (10 ml/kg) of ketamine (4.28 mg/ml), xylazine (0.86 mg/ml), and acepromazine (0.14 mg/ml) in saline. Periodontitis was induced by tying a 6–0 silk ligature (Fine Science Tools, BC, Canada; cat. no. 1802060) around the left maxillary second molar; the contralateral molar was left unligatured to serve as baseline control[35]. After 1 week, mice were euthanized, ligatures were recovered for microbial genomic DNA extraction, and maxillae were dissected for alveolar bone level measurement. In addition, gingival tissue around the maxillary second molar was collected for RNA extraction.

**Quantitative real-time PCR**. For bacterial quantification, microbial genomic DNAs were extracted from recovered ligatures. qPCR amplification was performed on a StepOnePlus™ Real-Time PCR System (Applied Biosystems, Foster City, CA). For quantification of bacteria amount, the reaction mixture (20 μl) contained TaqMan Fast Universal Master Mix (Applied Biosystems; cat. no. 4444557), microbial genomic DNA (2 μl), forward and reverse primers (Uni_152F and Uni_220R; 500 nM each), and probes (Uni_177P; 250 nM). Threshold cycle (CT) values were determined, and the log of colony-forming unit per ligature was calculated based on the standard curve. For quantification of NI1060, the reaction mixture (20 μl) contained Fast SYBR Green Master Mix (Applied Biosystems; cat. no. 4385612), microbial genomic DNA (2 μl), and forward and reverse primers (NI1060_F/ NI1060_R for NI1060; Uni_152F/Uni_220R for all bacteria; 500 nM each). The fold change of NI1060 percentage were calculated by the $2^{-\Delta\Delta CT}$ method ($\Delta\Delta CT = (CT_{NI1060/Gnat3^{-/-}} - CT_{bacteria/Gnat3^{-/-}}) - (CT_{NI1060/WT} - CT_{bacteria/WT})$).

For quantification of the expression levels of pro-inflammatory cytokines and AMPs, the reaction mixture (10 μl) contained Fast SYBR Green Master Mix, gingival cDNA (100 ng), and forward and reverse primers (500 nM each) for IL-1β, IL-6, IL-17, RANKL, Defb1, Defb2, Defb3, Camp, and β-actin. The expression level of target genes normalized with β-actin (presented in Fig. 5b) were calculated by $2^{-\Delta CT}$ method ($\Delta CT = CT_{Target} - CT_{\beta\text{-actin}}$). In addition, for relative mRNA expression calculation in ligature-induced periodontitis (presented in Fig. 3c, e), the expression levels of target genes in the left ligatured side were normalized by that in the right unligatured side. All primers are listed in Supplementary Table 3.

**Topical treatment with bitter Den.** WT and Gnat3$^{-/-}$ mice (8 weeks of age, $n =$ 6 for each group) had their gingivae topically treated with 1 mM Den in PBS or PBS alone as control, twice daily for 6 days (Fig. 5a). The maxillary gingival surface was brushed with a cotton swab soaked with Den or PBS. To study the effect of Den treatment on ligature-induced periodontitis, molar ligature was performed at day 0 on WT and Gnat3$^{-/-}$ mice aged 8 weeks ($n = 6$ for each group; Fig. 5c). At day 7, mice were euthanized to collect gingival tissues, maxillae, and ligatures for experiments indicated above.

**Quantification and statistical analysis.** For 16S rDNA sequencing data, statistical analyses were performed with I-Sangers online tools. The differences in beta diversity (revealed by PCA) within groups were compared with ANOSIM and Adonis; the alpha diversity data and genus-level microbial composition data were analyzed by Wilcoxon rank-sum test. Statistical analyses of other data were performed with the Prism software. All data were expressed as the mean, mean ± SEM, mean ± 95% confidence interval, or standard box plot as described in the figure legends. Multiple group comparisons were performed by one-way analysis of variance test followed by Tukey's test to identify differences between specific groups. For the NI1060 quantification data, two-group comparison was performed by Wilcoxon rank-sum test. Data were considered significantly different if the two-tailed $p$ value was <0.05. Sample sizes are given in the figure legends.

**Reporting summary.** Further information on research design is available in the Nature Research Reporting Summary linked to this article.

## Data availability

Microbiome sequencing data have been deposited in public database Sequence Read Archive (http://www.ncbi.nlm.nih.gov/Traces/sra) with accession nos. SRP126006 and SRP215896. The source data underlying Figs. 1a, c, d, 2b–j, 3b–e, 4a–e, and 5b, e, f and Supplementary Figs. 2A–C, 3A, 8A, 4A, B, 5A–D, and 6B–E are provided as a Source Data file. All other data sets generated during and/or analyzed during the current study are available from the corresponding author on reasonable request.

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

## Acknowledgements

The authors acknowledge helpful discussions from Dr. Liquan Huang and Dr. Hong Wang, Pou2f3−/− mice from Dr. Ichiro Matsumoto (Monell Chemical Senses Center), ChAT-GFP mice from Dr. Sukumar Vijayaraghavan (Univ. Colo. Denver), and bacterially produced AHLs, LasI, and EsaI from Dr. Mair Churchill (Univ. Colo. Denver). This work was supported by NIH-NIDCD grants R01 DC014105 (to R.F.M.), R03 DC012413 and R01DC016598 (to M.T.), and R01 DC013807 (to P.J.); the National Natural Science Foundation of China (81670978 to X. Zhou, 81771099 to X.X., 81600874 to J.H.), the Postdoctoral Foundation of Sichuan University Grant (2019SCU12021 to X. Zheng), and the Open Fund of the State Key Laboratory of Oral Diseases, Sichuan University (SKLOD2016OF03 to X. Zheng). Imaging was performed at the Monell Histology and Cellular Localization Core, which was supported, in part, by funding from NIH-NIDCD Core Grant P30 DC011735 (to R.F.M.) and National Science Foundation Grant DBI-0216310 (to Gary Beauchamp).

## Author contributions

Conceptualization: X. Zheng, M.T., X.X., X. Zhou, and R.F.M.; methodology: X. Zheng, M.T., and K.R.; formal analysis: X. Zheng and J.H.; investigation: X. Zheng, X.X., M.T., J.H., and X.P.; resources: M.T., P.J., and R.F.M.; data curation: J.H. and X.P.; writing original draft: X. Zheng, M.T., X.X., and R.F.M.; writing, review, and editing: M.T., P.J., X. Zhou, and R.F.M.; visualization: X. Zheng, X.X., and J.H.; supervision: M.T., X. Zhou, and R.F.M.; funding acquisition: X. Zheng, M.T., J.H., P.J., X.X., X. Zhou, and R.F.M.

## Competing interests

The authors declare no competing interests.

## Additional information

**Peer review information** *Nature Communication* thanks anonymous reviewers for their contributions to the peer review of this file. Peer review reports are available.

