## [Peer Review File · Nature Communications]

Reviewers' comments:

Reviewer #1 (Remarks to the Author):

The authors show a role for taste signaling molecules and gingival solitary chemosensory cells in the composition of the oral microbiota and alveolar bone loss. Overall the quality of the work is very good.

I have only minor comments:

1- The authors show mRNA expression of inflammatory cytokines comparing ligated/unligated wildtype (WT) and Gnat3 KO mice. Was the basal level of the controls already different between WT and KO? It would help if the authors show the fold increase of each control group separately. One cannot be sure if the basal levels of mRNA expression (unligated) are already higher in the knockouts - and if so, authors should discuss this.

2- The authors use methylene blue to demonstrate the levels of alveolar bone loss in the ligature-induced periodontitis. It would be better to show the bone loss by microCT. Methylene blue is only one parameter - the height of bone (distance from CEJ to ABC), but not the bone volume which should also be considered. It is possible that the Gnat3 KO mice could have less bone volume which would result in a lower amount of bone at the beginning. This would clarify if the KO mice might already have an inherent deficiency in bone volume.

3- Were mice cohoused before the in vivo experiments? Mice should be co-housed at the beginning of the in vivo experiments with the ligature as the effect of microbiota is being investigated. Since the authors show different microbial communities on the ligatures from WT vs. Gnat3^{-/-} mice which also differed in terms of beta diversity, it makes sense to cohoused the animals. Such an approach would be appropriate to make sure the effects that authors describe on the periodontal bone loss are actually due to the genetic deficiency and not due to naturally different microbiota.

Reviewer #2 (Remarks to the Author):

This paper reports an exciting and novel finding that specialized chemosensory cells of the gingiva utilize taste-receptor signaling to detect pathogenic bacteria and evoke a protective response by releasing antimicrobial peptides in mice. Absent these cells or their taste-signaling systems, mice show a tendency towards developing pathogenic gingival bacterial populations which lead to gingival recession and periodontitis.

Only a few minor issues should be resolved before this work would be ready for publication.

The authors mention the presence of GPR41 and 43, receptors for short chain fatty acids but not reported in taste buds, but do not mention whether they detect GPR120 and CD36 which are found in taste buds. One wonders whether the investigators also tested for these GPCRs.

The paper leaves unclear 2 important points: 1) Are any or all of the gSCCs innervated, and 2) do the gSCCs have an apical process reaching the surface of the epithelium? The latter point might be addressed by staining for villin. Also unclear is what is the nature of the gingival cells expressing Chat-driven GFP. Also the image showing this populations (Suppl Fig. 1C) is far too small to be useful.

Fig. 1B: Micrographs are too small and too dark to enable a reader to appreciate the situation of these cells. These images also lack labels giving directionality, e.g. which is the side facing the tooth?

Line 171-173: In describing the overgrowth of bacteria, the authors state:

To determine if overgrowth...was due to diminished..." This overstates what was tested. The experiment does not show causality but merely correlation.

Fig. 3G, although attractive, is not very informative to an average reader. Some more description in the caption may be helpful to guide the reader to the import of the figure. Also the use of green lines

to denote negative correlations would seem opposite from a typical reader's presumptions about the implied meanings of red and green. Were the same species being compared across conditions? Were relatively rare species eliminated from this graphical presentation? (They should have been).

Fig 4B: The crucial information as to which Def is being measured is hidden in the small vertical text alongside of the Y axis of the graphs in panel B. Recommend putting the name of the Def being measured in larger lettering in the upper R corner of each panel and merely label the axis as Def/ActB (or Camp) in each panel.

Text Line 215: should. read: "for denatonium-induced protection..."

Line 218: Discussion: Backwards written is this first sentence. It should begin with the subject.

Line 254: The word "express" is over-used here. Tissues can express molecular features, but they don't express a cell type.

The discussion of TAS2R38 in airways seems excessive and dilutes the importance of the relationship of TAS2R38 to dental conditions.

Please give institutional affiliations of investigators who have provided mice or other reagents.

To the Editor & Reviewers:

The authors thank the reviewers for their careful reading of our manuscript and their thoughtful comments. Below we provide point by point responses to each of the reviewer's comments. For specific text changes we refer to the tracked version of the manuscript.

Response to Reviewer #1:

Reviewer #1 (Remarks to the Author):

The authors show a role for taste signaling molecules and gingival solitary chemosensory cells in the composition of the oral microbiota and alveolar bone loss. Overall the quality of the work is very good.

I have only minor comments:

1- The authors show mRNA expression of inflammatory cytokines comparing ligated/unligated wildtype (WT) and Gnat3 KO mice. Was the basal level of the controls already different between WT and KO? It would help if the authors show the fold increase of each control group separately. One cannot be sure if the basal levels of mRNA expression (unligated) are already higher in the knockouts - and if so, authors should discuss this.

Response: To address these points we now show the absolute levels of cytokine mRNAs from the control unligated side from both the WT and Gnat3^{-/-} groups (see new Extended Data Fig 4). These data show that basal levels of mRNAs for Il1b, Il6, Il17, Rankl, Defb1, Defb2 and Camp are similar for WT and gustducin KO mice. It is only for the basal Defb3 absolute mRNA levels that there is a statistically significant decrease (~2-fold) in the KO vs. WT mice. The 2-fold difference in Defb3 mRNA levels is comparable to that seen with relative mRNA levels in Fig 3 E. Thus, the absence of gustducin reduces the basal level of Defb3 in gingival tissue from Gnat3 KO mice, and impairs the upregulation of Defb3 in the ligature-induced periodontitis model. This is noted in the text (tracked lines 168-170, 180-182) and touched on in the Discussion (tracked lines 303-5).

2- The authors use methylene blue to demonstrate the levels of alveolar bone loss in the ligature-induced periodontitis. It would be better to show the bone loss by microCT. Methylene blue is only one parameter - the height of bone (distance from CEJ to ABC), but not the bone volume which should also be considered. It is possible that the Gnat3 KO mice could have less bone volume which would result in a lower amount of bone at the beginning. This would clarify if the KO mice might already have an inherent deficiency in bone volume.

Response: To determine if Gnat3 KO mice have less bone volume at the beginning of the experimental manipulation we performed microCT on unligated WT and Gnat3 KO mice of 8 and 16 weeks of age (the same ages used in the naturally-occurring periodontitis experiments of Fig. 2). MicroCT was used to measure the following alveolar bone parameters: bone

volume/total volume, trabecular number and trabecular thickness. The bone volumes in both WT and KO mice at both ages were similar (see new panels Fig 2 C-E and tracked lines 135-137), indicating that the KO mice do not have decreased bone volume or an inherent bone deficiency at 8 or 16 weeks of age. This indicates that the differences measured by methylene blue in WT vs. KO mice for the naturally-occurring alveolar bone loss (Fig 2 A,B) or ligature-induced periodontitis (Fig 3 A,B) did not arise from different starting baselines.

3- Were mice cohoused before the in vivo experiments? Mice should be co-housed at the beginning of the in vivo experiments with the ligature as the effect of microbiota is being investigated. Since the authors show different microbial communities on the ligatures from WT vs. Gnat3^{-/-} mice which also differed in terms of beta diversity, it makes sense to cohoused the animals. Such an approach would be appropriate to make sure the effects that authors describe on the periodontal bone loss are actually due to the genetic deficiency and not due to naturally different microbiota.

Response: Mice were not cohoused before the in vivo ligature experiments, although they were divided over several cages to dilute cage effects. To our knowledge, the effects of cohousing on the oral microbiome are not as dominating as for cohousing's effects on the gut microbiome. Also, please note that the micro-CT data showed similar bone density for WT and Gnat3 KO mice at 8 weeks, indicating that prior to the placement of the ligature the genetic deficiency of Gnat3 *per se* does not result in significant periodontal bone loss, but after introducing the ligature alterations in the microbiota and/or differences in innate immune responses between WT and Gnat3 KO may lead to a more susceptible host with increased ligature-induced periodontal bone loss. We do plan for future experiments to test the effects on the gingival microbiome of co-housing, particularly in older mice (>1 year), and to examine the effects of microbiome transplantation from WT and Gnat3 KO using germ-free mice. However, we feel that these experiments are beyond the scope of the present work.

Response to Reviewer #2:

Reviewer #2 (Remarks to the Author):

This paper reports an exciting and novel finding that specialized chemoresponsive cells of the gingiva utilize taste-receptor signaling to detect pathogenic bacteria and evoke a protective response by releasing antimicrobial peptides in mice. Absent these cells or their taste-signaling systems, mice show a tendency towards developing pathogenic gingival bacterial populations which lead to gingival recession and periodontitis.

Only a few minor issues should be resolved before this work would be ready for publication:

1- The authors mention the presence of GPR41 and 43, receptors for short chain fatty acids but not reported in taste buds, but do not mention whether they detect GPR120 and CD36 which are found in taste buds. One wonders whether the investigators also tested for these GPCRs.

Response: By RT-PCR we find that both GPR120 and CD36 are expressed in both taste and gingival tissues. By the same technique we find that GPR41 and GPR43 are expressed in gingival tissue but not in taste tissue. This is now noted in the text (tracked line 86) and shown in newly added Extended Data Fig 1B. Interestingly, both GPR41 and GPR43 are co-expressed with GNAT3 in enteroendocrine cells of the colon (Li et al., AJP, 2013). These short chain fatty acid receptors may function like TAS2Rs in gingival SCCs to detect bacterial metabolites. For future studies it would be of great interest to determine which gingival SCC receptors interact with which bacterial-derived signaling molecules to promote growth of commensals over pathogens in regulating the composition of the gingival microbiome.

2- The paper leaves unclear 2 important points: 1) Are any or all of the gSCCs innervated, and 2) do the gSCCs have an apical process reaching the surface of the epithelium? The latter point might be addressed by staining for villin. Also unclear is what is the nature of the gingival cells expressing Chat-driven GFP. Also the image showing this populations (Suppl Fig. 1C) is far too small to be useful.

Response: We examined gSCCs with antibodies against villin, PGP9.5 and acetylated tubulin; unfortunately, our procedures for decalcifying the tissue are incompatible with these particular antibodies, so the questions regarding gSCC innervation and their processes reaching the surface remain unanswered. Some of our other immuno-stainings showed gSCCs with an apical process at the surface but the direction is parallel to instead of toward the surface (see Fig. 1B and Extended Data Fig 1C); more frequently we observed SCCs with apical processes not at the surface but deeper within the epithelial tissue. We agree that the nature of the Chat-positive cells located in the deeper part of the oral epithelium is unclear at present. In contrast to nasal SCCs, gingival SCCs do not express Chat and so it was somewhat surprising to find these Chat-positive non-SCC cells deeper in the epithelium. Presumably gSCCs do not use acetylcholine as a neurotransmitter and instead function locally with the immune system to keep pathogenic microbes from invading the teeth or alveolar bone. Extended Data Fig 1C was increased in size to be more accessible to the reader.

3- Fig. 1B: Micrographs are stop small and too dark to enable a reader to appreciate the situation of these cells. These images also lack labels giving directionality, e.g. which is the side facing the tooth?

Response: We adjusted the figure accordingly.

4- Line 171-173: In describing the overgrowth of bacteria, the authors state: "To determine if overgrowth...was due to diminished..." This overstates what was tested. The experiment does not show causality but merely correlation.

Response: The text was rewritten to indicate correlation not causation.

5- Fig. 3G, although attractive, is not very informative to an average reader. Some more description in the caption may be helpful to guide the reader to the import of the figure. Also

the use of green lines to denote negative correlations would seem opposite from a typical reader's presumptions about the implied meanings of red and green. Were the same species being compared across conditions? Were relatively rare species eliminated from this graphical presentation? (They should have been).

Response: We changed the color of lines indicating correlations between OTUs in Fig. 3G, and added more explanatory text in the figure legend and in the Materials and Methods to make it clearer to readers. The OTUs (each circle) in WT or Gnat3^{-/-} groups were not the same: the 50 most abundant OTUs in each group were analyzed, but only the OTUs that correlated are shown in the figure.

6- Fig 4B: The crucial information as to which Def is being measured is hidden in the small vertical text alongside of the Y axis of the graphs in panel B. Recommend putting the name of the Def being measured in larger lettering in the upper R corner of each panel and merely label the axis as Def/ActB (or Camp) in each panel.

Response: We adjusted the figure accordingly.

7- Text Line 215: should. read: "for denatonium-induced protection..."

Line 218: Discussion: Backwards written is this first sentence. It should begin with the subject.

Line 254: The word "express" is over-used here. Tissues can express molecular features, but they don't express a cell type.

The discussion of TAS2R38 in airways seems excessive and dilutes the importance of the relationship of TAS2R38 to dental conditions.

Please give institutional affiliations of investigators who have provided mice or other reagents.

Response: The manuscript was rewritten accordingly.

REVIEWERS' COMMENTS:

Reviewer #2 (Remarks to the Author):

The revisions made in response to the initial reviews are adequate.